# Authentication of Laying Hen Housing Systems Based on Egg Yolk Using ^1^H NMR Spectroscopy and Machine Learning

**DOI:** 10.3390/foods13071098

**Published:** 2024-04-03

**Authors:** Greta Bischof, Edwin Januschewski, Andreas Juadjur

**Affiliations:** Chemical Analytics, German Institute of Food Technologies (DIL e.V.), Prof.-v.-Klitzing-Str. 7, 49610 Quakenbrück, Germanya.juadjur@dil-ev.de (A.J.)

**Keywords:** chicken egg, husbandry, organic, classification, discriminant analysis, support vector machine, artificial neural network, random forest, k-nearest neighbor, partial least square

## Abstract

(1) Background: The authenticity of eggs in relation to the housing system of laying hens is susceptible to food fraud due to the potential for egg mislabeling. (2) Methods: A total of 4188 egg yolks, obtained from four different breeds of laying hens housed in colony cage, barn, free-range, and organic systems, were analyzed using ^1^H NMR spectroscopy. The data of the resulting ^1^H NMR spectra were used for different machine learning methods to build classification models for the four housing systems. (3) Results: The comparison of the seven computed models showed that the support vector machine (SVM) model gave the best results with a cross-validation accuracy of 98.5%. The test of classification models with eggs from supermarkets showed that only a maximum of 62.8% of samples were classified according to the housing system labeled on the eggs. (4) Conclusion: The classification models developed in this study included the largest sample size compared to the literature. The SVM model is most suitable for evaluating ^1^H NMR data in terms of the hen housing system. The test with supermarket samples showed that more authentic samples to analyze influencing factors such as breed, feeding, and housing changes are required.

## 1. Introduction

Organically produced animal foods are more expensive to produce due to the stricter regulations on feeding and housing systems related to animal welfare, making them a target for food fraud [1,2]. Food fraud can be counteracted by authenticating organically produced food. Therefore, analytical methods are required for the authentication of animal food, e.g., in relation to animal species, breed, feeding, or housing system [3,4]. For example, conventionally produced hen’s eggs can easily be labeled as organically produced eggs due to their identical external appearance. The production of eggs and the husbandry of laying hens are subject to a number of EU regulations and directives [1,5,6], which define the minimum requirements for the different housing systems, in particular for organic farming. In Germany, eggs from laying hens are labeled with a code that symbolizes the housing system (colony cage: 3; barn: 2; free-range: 1; organic: 0) and the farm where the eggs were produced. The colony cage system is currently approved in Germany until 2025.

Several studies have focused on the classification of the housing system of laying hens based on the metabolome of the eggs. Ackermann et al. [7] reported a ^1^H NMR method combined with linear discriminant analysis (LDA) to differentiate between organically and conventionally produced eggs (*n* = 344 samples). Puertas and Vázquez [8] determined the housing systems (cage, barn, free-range, and organic) by UV-Vis-NIR spectroscopic analyses of egg yolk lipid extract (*n* = 48 samples) [8] or egg plasma (*n* = 84 samples) [9] and support vector machine (SVM), quadratic discriminant analysis (QDA), and LDA. In addition, Hajjar et al. [10] analyzed the triacylglycerol extract of egg yolk (*n* = 34 samples) using ^1^H NMR spectroscopy and classified the housing system by LDA and canonical discriminant analysis. Cardoso et al. [11] showed the discrimination of barn and free-range eggs (*n* = 48 samples) by ^1^H NMR spectroscopy and partial least square discriminant analysis (PLS-DA). Chin et al. [12] reported an LC-MS/MS method to analyze lipid extracts from egg yolks (*n* = 357 samples) from cage, barn, and free-range systems in combination with the machine learning methods orthogonal projection in latent structures discriminant analysis (OPLS-DA) and SVM. The OPLS-DA and SVM models showed accuracies between 77 and 96% and 80 and 97%, respectively, depending on the class [12]. Furthermore, Lösel et al. [13] analyzed 270 eggs from conventional and organic systems with LC-ESI-IM-qTOF-MS and FT-IR spectroscopy in combination with a random forest (RF) calculation and reported an accuracy of 96.3%. Kopec and Abramczyk [14] showed a PLS-DA model based on Raman spectra of 40 eggs from cage, barn, free-range, and organic systems with a sensitivity and specificity of validation between 0.897 and 1.0.

To the best of our knowledge, this is the first study to authenticate the housing system of laying hens with a sample size *n* > 1000 and to develop seven machine learning methods and apply the models to egg samples purchased from supermarkets. 

We hypothesized that it is possible to authenticate the housing system of laying hens by ^1^H NMR analysis of a large number of authentic egg yolk samples and an appropriate machine learning model. 

## 2. Materials and Methods

### 2.1. Raw Material

The egg samples were collected from the farms of a German egg producer. The eggs were from laying hens from the four different housing systems (colony cage, barn, free-range, and organic) and four different breeds (Lohmann Selected Leghorn, Dekalb, Lohmann Brown, and Sandys). The laying period of the hens was separated into four groups: <25 weeks, 26–35 weeks, 36–55 weeks, and >56 weeks. Eggs were collected throughout the entire laying period of the hens (20 to 94 weeks), with every effort being made to ensure that at least one sample from each combination of age group, breed, and housing system was available. Further information is presented in Appendix A. A total of 472 eggs were collected from colony cages, 1200 eggs from barn, 1192 eggs from free-range and 1324 eggs from organic housing systems, resulting in a total amount of 4188 eggs used for model development. In addition, 290 eggs labeled as barn, free-range, or organic housing systems were purchased from supermarkets and discounters (Appendix A).

### 2.2. Sample Preparation for ^1^H NMR Analysis

The sample preparation method was conducted according to the method of Ackermann et al. [7] with some modifications (Figure 1). After the separation of egg yolk from egg white, 3 g of each egg yolk was lyophilized at −80 °C and 0.09 mbar for 18 h (Alpha 2-4 LDplus, Martin Christ Gefriertrocknungsanlagen GmbH, Osterode am Harz, Germany). A total of 100 mg of sample was homogenized (Bead Ruptor Elite Bead Mill Homogenizer; OMNI International, Kennesaw, GA, USA) with 938 µL of a chloroform–methanol–water mixture (10:5:1; *v*/*v*/*v*, CHCl_3_, ≥99.8%, VWR International, Philadelphia, PA, USA; MeOH, ≥99.9%, Merck KGaA, Darmstadt, Germany; ultra-pure water, Milli-Q Organex-Q System, Merck, Millipore, Darmstadt, Germany) and two stainless steel beads in two cycles at 3.1 m/s for 30 s. The samples were centrifuged (Hettich Universal 320 R, Hettich GmbH & Co. KG, Tuttlingen, Germany) at 18,620× *g* and 4 °C for 20 min. After the addition of 62.5 µL of 0.05 M NaCl solution (≥99.5%, Applichem GmbH, Darmstadt, Germany), the samples were centrifuged at 18,620× *g* and 4 °C for 3 min. An amount of 400 µL of the lower phase was dried under nitrogen at 50 °C for 60 min (Evaporator EVA-EC1-24-S, VLM GmbH, Bielefeld, Germany). The dried samples were dissolved in 400 µL CDCl_3_-*d1* (99.8%D; with 0.03% tetramethylsilane; Carl Roth GmbH & Co. KG, Karlsruhe, Germany) and dried under nitrogen at 50 °C for 5 h. After dissolution of the dried samples in 800 µL CDCl_3_-MeOD mixture (3:2; *v*/*v*; 99.8%D MeOD-*d4* with 0.03% tetramethylsilane; Acros Organics B.V.B.A., Geel, Belgium) plus 3 mM 1,3,5-trimethoxybenzene (≥99%; Sigma Aldrich, St. Louis, MO, USA), 600 µL were transferred into a 5 mm NMR tube (Deutero GmbH, Kastellaun, Germany).

### 2.3. ^1^H NMR Spectroscopy

The samples were analyzed using a 400 MHz Ascend III NMR spectrometer (Bruker Biospin GmbH, Ettlingen, Germany) with the following measurement parameters: pulse program, zg; temperature, 298 K; spectral width, 8223 Hz; number of points, 65 k; number of scans, 128; number of dummy scans, 4; acquisition time, 3.9 s; and relaxation delay, 6.0 s. Phase and baseline corrections were performed automatically using TopSpin 3.6.5 (Bruker Biospin GmbH).

### 2.4. Data Analysis

Data analysis was performed using Matlab R2018a (The Mathworks, Natick, MA, USA) and RStudio 2023.06.0 421 (Posit PBC, Boston, MA, USA) based on the R 4.3.1 software. The ^1^H NMR spectra were scaled using the signal of trimethoxybenzene as an internal standard compound and conventionally bucketed into 682 buckets with a size of 0.01 ppm. The signals from MeOD and CDCl_3_ were excluded. The machine learning models (LDA, QDA, PLS-DA, SVM, RF, k-nearest neighbor (kNN), and artificial neural network (ANN)) were performed based on the buckets of ^1^H NMR spectra. Since three out of four classes (housing system) were approximately equal in size, and only one class (colony cage) contained fewer samples, the data were not adjusted for imbalance. This avoids negative effects such as replication of minor effects or loss of information. In the case of LDA, QDA, SVM, RF, kNN, and ANN, a principal component analysis (PCA) was computed beforehand, and 134 principal components were used for these model calculations. A radial basis kernel function with degree = 3 and cost = 1 was used for SVM computation. The parameters ntree (number of trees) and mtry (number of variables) for the RF model, number of PLS components for PLS-DA, k (number of neighbors) for kNN, number of hidden layers, and number of neurons in each hidden layer for ANN were optimized in terms of model accuracy and set to ntree = 800, mtry = 8, number of PLS components = 13, k = 5, number of hidden layers = 2, and number of neurons in each hidden layer = 5.

In general, the data set was randomly divided into a training set (80% of the data) and a test set (20% of the data). The training set was used to compute the different models and the test set for validation. Additionally, all model calculations were validated by k-fold cross-validation with k = 10. Accuracy, sensitivity, specificity, precision, number of misclassifications (NMC), and area under the curve (AUC) of the receiver operating characteristic (ROC) curve were calculated for the model, the cross-validated model, and the test set fit. A permutation test with 1000 permutations was conducted for each model, and the *p*-value for accuracy, NMC, and AUC were reported. These tests and parameters provide information on the quality of the models and on the problem of the models with imbalanced data. The equations are provided in the Appendix A.

## 3. Results and Discussion

### 3.1. Model Computing Using Multiple Machine Learning Methods

The egg yolks of 4188 eggs from the four housing systems (colony cage, barn, free-range, and organic) were separated, extracted, and measured using ^1^H NMR spectroscopy (Figure 1). The ^1^H NMR spectra (Figure 2) are the basis for the calculation of the different machine learning models. Literature has shown that several classification methods are applicable to NMR data. Seven machine learning methods (LDA, QDA, SVM, PLS-DA, RF, kNN, ANN) were selected to develop the classification models. In the case of LDA and QDA, the between-class variances are maximized, and the within-class variances are minimized to increase distinctiveness. Unlike QDA, LDA assumes equal covariance matrices between classes. In addition, the LDA has a linear decision surface, while the QDA has a nonlinear decision surface [15,16]. The SVM calculates hyperplanes to discriminate different classes, both linearly and non-linearly, by combining with the kernel function [17]. The PLS-DA is a supervised linear classifier that uses PCAs on x and y terms. Direct relationships to the classes and the analyzed metabolome can be established by calculating loadings and variables’ importance in projection (VIP) scores [18,19,20]. In the case of RF, multiple low-correlated decision trees are constructed based on random splits of the data. The predicted class is the class selected by the most trees, thus minimizing the error of using only one decision tree. RF can handle large data sets with high dimensionality and large sample size [21,22]. The kNN classification is based on the classes of a certain number (k) of the nearest data points (neighbors). The kNN algorithm is easy to apply and can be helpful in cases where other machine learning methods fail [23]. The ANN is a complex method that consists of input, hidden, and output layers with a defined number of artificial neurons depending on the application [24].

The validation results of the different models are shown in Table 1 and Table 2. The accuracies of fit, cross-validation, and test set were close to each other for the classification models, indicating that there were no overfitting models. The *p*-values of the permutation test showed that the computed models with permuted class labels had a significantly poorer classification performance than the correct models. This indicates that the models with correct labels were useful in predicting the housing system. The models with the highest accuracy of fit (99.9%), the highest accuracy of cross-validation (97.8%; 98.5%), and the lowest NMC (3–7; 1–5) were QDA and SVM. The QDA model showed wider ranges in sensitivity (0.879–0.994), specificity (0.979–0.999), precision (0.949–0.997) and AUC (0.939–0.997) of cross-validation than the SVM model (sensitivity: 0.953–1.000; specificity: 0.993–0.999; precision: 0.983–0.989; AUC: 0.976–0.996), but the QDA is the only model where the AUC for the class colony cage in the permutation test was as high as the AUC of the normal model. This indicates that the classification of the class colony cage was random. QDA and SVM are both classifiers that are not based on a linear decision surface like LDA. QDA can be based on a quadratic function or curve, while SVM is based on several kernel functions (e.g., polynomial, radial, etc.) [15,17]. A different characteristic of multi-class applications between SVM and LDA/QDA is that LDA/QDA computation focuses on very different classes, while SVM computation focuses on closer classes [25]. This may explain the weakness of QDA in classifying colony cage samples. Furthermore, the class colony cage is unrepresentative in terms of sample size (547 samples) compared to the other classes (>1200 samples) due to the reduced use of colony cage in Germany, which could explain the poorer classification performance of this class. This is reflected in the lower sensitivity of the colony cage class in all models compared to the other three classes.

The accuracy of the test set samples was the highest for the QDA and SVM models (98.6% and 99.0%, respectively) compared to the other models. Chin et al. [12] showed that their SVM model provided better classification results than the OPLS-DA model when using MS/MS spectra of eggs from conventional housing systems. In contrast to the results of this study, the SVM model by Puertas and Vázquez [8] and Puertas et al. [9] showed worse classification results than LDA and QDA. These differences may be explained by the use of UV-Vis and NIR data to calculate the models. In general, the SVM algorithm could be useful for authenticating food samples based on NMR spectra [22,26,27,28]. For example, Cui et al. [26] demonstrated that their SVM model had the best classification results compared to LDA, PLS-DA and RF to authenticate the geographical origin of *Zanthoxylum bungeanum* extract samples. As another example, Nyitrainé Sárdy et al. [28] compared LDA, ANN, SVM, and RF models to classify the region and variety of wine based on ^1^H NMR spectra and showed that the SVM and RF models had the best classification performance.

The results of LDA compared to QDA showed a poorer classification model, which is consistent with observations noted in the literature regarding eggs [8,9] or other foods [26,28] and could be an indicator that a linear decision surface is less applicable for these samples. Hajjar et al. [10] and Ackermann et al. [7] used ^1^H NMR spectra of conventionally and organically produced eggs to compute a good classification model based on LDA. This may be due to the fact that there were only two classes (conventional vs. organic), which may be easier to separate by the LDA model than the four housing systems.

The model with the lowest accuracies was the PLS-DA, with 85.6% (fit), 84.7% (CV), and 84.6% (test set). In the literature, PLS-DA and OPLS-DA have often been used for food classification approaches based on data from NMR spectra [29,30,31,32]. In relation to eggs, Cardoso et al. [11] reported a PLS-DA model to classify barn and free-range eggs using the ^1^H NMR spectra. The RF, kNN, and ANN models showed poorer classification results than SVM, QDA, and LDA but better than PLS-DA.

Therefore, the SVM model is the best model in this study with a cross-validation accuracy of 98.5% to predict the housing system of laying hens using ^1^H NMR spectra. Compared to the literature, Jiménez-Carvelo et al. [33] summarized that the SVM algorithm is suitable for several food classification/authentication approaches, for example, in the case of honey, meat, milk, plant products and oils.

### 3.2. Prediction of Housing System from Supermarket Samples

To evaluate the models’ predictive performance in a practical setting, 290 egg samples labeled ‘0’ (organic), ‘1’ (free-range), and ‘2’ (barn) were purchased from supermarkets or discount stores. Eggs labeled ‘3’ (colony cage) were not commercially available. The outcomes of the model predictions are detailed in Table 2 and Table 3.

Prediction of the supermarket sample set was performed with all seven models to compare prediction results and provide information on model weakness or sample outliers. For samples V-001, V-002, V-003, V-004, V-005, V-007, V-013, V-017, V-018, V-026, and V-027, (almost) all models correctly predicted the labeled housing system for (almost) all eggs. This observation indicates that the eggs were correctly labeled with a very high statistical certainty. If all or a large proportion of the eggs in a sample cannot be predicted as labeled by all or almost all models then it is reasonable to assume that this sample is an outlier. Mislabeled samples, as well as samples that differ greatly from the training set, for example, due to a different breed, are possible outliers. This was observed for samples V-006, V-009, V-014, V-016, V-019, V-020, V-025, and V-028 (Table 3). All models classified only 45.9% (kNN) to 62.8% (LDA) of the samples into the class printed on the shell (Table 2). The best model (SVM) mentioned above predicted 52.8% of the samples according to the labeled housing system. This observation may be due to the small sample size (4188 eggs) in relation to the number of laying hens in Germany (43.7 million hens in the year 2022 [34]) and/or due to a higher number of mislabeled eggs than expected. The prediction results (Table 3) demonstrated that most of the eggs labeled as barn eggs were misclassified. One reason could be that laying hens from free-range farming have to be housed indoors when sudden external influences occur (e.g., avian influenza risk area) [35], meaning the housing system is temporarily comparable to barn housing. In this case, the eggs from these laying hens had to be labeled as barn eggs if they were kept indoors for more than 12 weeks [36]. Since a new regulation came into effect in November 2023, such eggs can still be labeled as free-range eggs today [5]. However, the eggs used in this study were collected before the new regulation came into effect. In addition, the sample size for barn eggs may still be too small to provide a representative sample. In Germany, the barn system contains more than twice as many laying hens as the free-range system and more than five times as many laying hens as the organic system. It can therefore be concluded that the population of barn eggs is significantly larger and is therefore less well represented by the same number of samples than in the other systems. Thus, the authentication model developed here may be improved using a larger number of samples, in particular from barn eggs, which will provide a more representative sample.

Additionally, the age of the laying hens may impact the metabolome of the eggs and thus be an influencing factor. It has been reported in the literature that the fatty acid pattern of the egg yolk is influenced by the age of the laying hens [37,38,39]. In this study, eggs from laying hens were collected throughout the laying period to minimize the influence of age in the machine learning models. Furthermore, the breed of the laying hens could be another influencing factor. Hejdysz et al. [40] analyzed eggs from 14 different breeds and reported significant differences in fatty acid profiles and cholesterol levels. There are further studies reporting differences in the metabolome of eggs from different breeds [10,41,42,43]. In the authentication model developed in this study, eggs from four different breeds were considered in order to include the variation between these breeds in the model and to be able to determine the housing system of laying hens despite this variation. This should make it possible to authenticate the housing system for laying hens regardless of their breed. Therefore, it is possible that the eggs collected in supermarkets and discounters were from breeds that were not included in the collection of authentic samples and therefore not included in the machine learning models. Furthermore, the classification models found in the literature were based on 34–357 samples, and half of the studies did not ensure the authenticity of the samples. Compared to the literature, this study presents the largest classification models using authentic egg samples.

Improving the authentication model to predict unknown samples requires adding more authentic samples. These authentic samples should include samples from other breeds or breed hybrids, depending on the breeds used in egg production, and from the four housing systems, especially the barn systems. In addition, studies on feeding and its effect on the metabolome of eggs and, thus, on the prediction of the authentication model will be helpful. The machine learning model may be useful to authenticate eggs with and without the shell to control the labeling or to determine the housing system, for example, in foods where eggs were used in production (e.g., egg noodles).

## 4. Conclusions

This is the first study with an authentication model for eggs from laying hens with respect to their farming method based on the data of ^1^H NMR spectra and a very large number of samples. It is also the first study to test their authentication model with samples from the supermarket. Firstly, the SVM model was the best model to authenticate egg samples compared to the LDA, QDA, PLS-DA, RF, kNN, and ANN models. Secondly, the model test with supermarket samples demonstrated that the prediction of the housing system needs further research in terms of influencing factors such as breed, feeding, and change of housing system (outdoor to indoor). Therefore, a large sample size is necessary to analyze and find influencing factors. It is also useful to test authentication models with real samples. To improve the model and adapt it to the new regulations, more authentic samples will be collected and added in the future.

## Figures and Tables

**Figure 1 foods-13-01098-f001:**
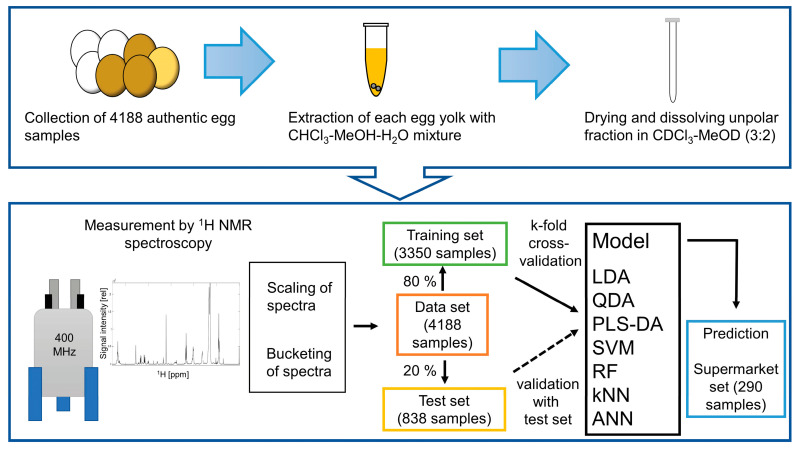
Overview of the preparation, processing, and calculation steps.

**Figure 2 foods-13-01098-f002:**
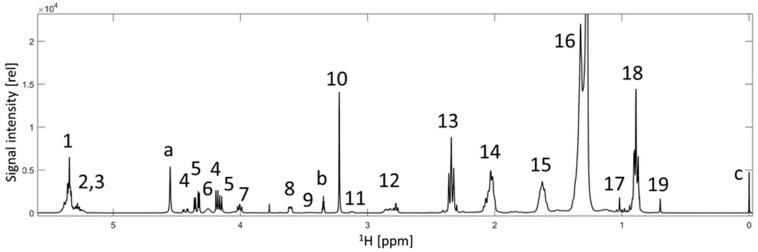
^1^H NMR spectrum of an egg yolk extract in CDCl_3_-*d1*/MeOD-*d4* (3:2). The relative signal intensity was plotted against the chemical shift from 0 to 5.5 ppm. ^1^H NMR spectrum was measured by a 400 MHz spectrometer. Signals related to triacylglycerides and the fatty acids: 1. vinyl hydrogen of all unsaturated fatty acids, 2, 5. Glycerol backbone, 14. allylic-methylene hydrogen of all unsaturated fatty acids, 12. CH_2_-bis-allylic hydrogen of polyunsaturated fatty acids (ω-3 and ω-6), 13. α-carbonyl methylene group, 15. methylene group at carbonyl β-position, 16. ethyl group, 18. methyl group of fatty acids. Signals related to phospholipids: 3, 4, 7. glycerol backbone, 6, 8, 10. phosphatidylcholine, 7 and 11. phosphatidylethanolamine. 19, 17, 19. cholesterol. Signals related to solvents or standard compounds: a,b. methanol, c. tetramethylsilane. The assignment of the signals to the components was carried out by comparison with Ackermann et al. [7] and self-measured standard components.

**Table 1 foods-13-01098-t001:** Results of fit and cross-validation of the models LDA, QDA, PLS-DA, SVM, RF, kNN, and ANN.

Model Type	Fit						k-Fold Cross-Validation (k = 10)					
	Accuracy	Sensitivity	Specificity	Precision	AUC ^1^	NMC ^2^	Accuracy	Sensitivity	Specificity	Precision	AUC ^1^	NMC ^2^
LDA	0.980	0.969	0.985	0.961	0.977	67	0.972	0.961	0.978	0.944	0.969	9
		0.979	0.990	0.975	0.985			0.973	0.990	0.974	0.981	
		0.956	0.998	0.984	0.977			0.928	0.995	0.962	0.961	
		0.999	1.000	1.000	1.000			0.998	0.999	0.999	0.999	
QDA	0.999	0.999	0.999	0.998	0.999	3	0.978	0.993	0.979	0.949	0.986	7
		0.998	1.000	1.000	0.999			0.985	0.992	0.980	0.989	
		1.000	0.999	0.997	1.000			0.879	0.999	0.994	0.939	
		1.000	0.999	1.000	1.000			0.994	0.999	0.997	0.997	
PLS-DA	0.856	0.822	0.955	0.876	0.888	168	0.847	0.804	0.953	0.869	0.878	182
		0.623	0.980	0.943	0.908			0.831	0.976	0.933	0.903	
		0.989	0.998	0.972	0.810			0.605	0.998	0.971	0.801	
		0.856	0.987	0.973	0.988			0.988	0.984	0.967	0.986	
SVM	0.999	1.000	0.999	0.999	1.000	1	0.985	0.978	0.994	0.985	0.986	5
		1.000	1.000	1.000	1.000			0.987	0.993	0.983	0.990	
		0.997	1.000	1.000	0.999			0.953	0.999	0.989	0.976	
		1.000	1.000	1.000	1.000			1.000	0.993	0.984	0.996	
RF	0.949	0.950	0.976	0.938	0.960	172	0.946	0.936	0.980	0.950	0.959	18
		0.957	0.980	0.949	0.966			0.943	0.982	0.954	0.962	
		0.939	0.985	0.881	0.937			0.879	0.993	0.947	0.936	
		0.943	0.992	0.982	0.977			0.982	0.968	0.934	0.975	
kNN	0.978	0.970	0.990	0.974	0.980	74	0.958	0.942	0.982	0.954	0.962	14
		0.990	0.988	0.971	0.989			0.977	0.979	0.949	0.978	
		0.946	0.997	0.976	0.971			0.899	0.993	0.943	0.946	
		0.986	0.994	0.988	0.990			0.975	0.988	0.974	0.981	
ANN	0.999	1.000	0.999	0.998	1.000	5	0.954	0.935	0.983	0.955	0.959	15
		0.997	1.000	1.000	0.998			0.959	0.985	0.962	0.972	
		0.995	1.000	1.000	0.997			0.920	0.990	0.920	0.955	
		1.000	1.000	1.000	1.000			0.978	0.996	0.991	0.987	

LDA—linear discriminant analysis, QDA—quadratic linear discriminant analysis, PLS-DA—partial least square discriminant analysis, SVM—support vector machine, RF—random forest, kNN—k-nearest neighbor, ANN—artificial neuronal network. Listed as barn, free-range, colony cage, and organic housing system. ^1^ AUC—area under the ROC (receiver operating characteristic) curve; ^2^ NMC—number of misclassifications.

**Table 2 foods-13-01098-t002:** Results of test set fit, permutation test, and purchased sample set for each computed model.

Model Type	Test Set						Permutation *p*-Value			Purchased Set
	Accuracy	Sensitivity	Specificity	Precision	AUC ^1^	NMC ^2^	Accuracy	AUC ^1^	NMC ^2^	Accuracy
LDA	0.973	0.989	0.969	0.935	0.979	23	<0.001	<0.001	<0.001	0.628
		0.945	0.993	0.982	0.969					
		0.918	1.000	1.000	0.959					
		1.000	0.998	0.996	0.999					
QDA	0.986	0.996	0.984	0.967	0.990	12	<0.001	<0.001	<0.001	0.569
		0.983	0.997	0.991	0.990			<0.001		
		0.918	1.000	1.000	0.959			0.999		
		1.000	0.998	0.996	0.999			<0.001		
PLS-DA	0.846	0.785	0.950	0.876	0.868	44	<0.001	<0.001	<0.001	0.552
		0.838	0.975	0.929	0.907					
		0.600	1.000	1.000	0.800					
		0.996	0.991	0.981	0.994					
SVM	0.990	1.000	0.988	0.974	0.994	8	<0.001	<0.001	<0.001	0.528
		0.974	1.000	1.000	0.987					
		0.976	1.000	1.000	0.988					
		1.000	0.998	0.996	0.999					
RF	0.942	0.931	0.974	0.942	0.953	49	<0.001	<0.001	<0.001	0.514
		0.953	0.964	0.911	0.958					
		0.847	0.997	0.973	0.922					
		0.973	0.983	0.962	0.978					
kNN	0.956	0.969	0.978	0.950	0.968	37	<0.001	<0.001	<0.001	0.459
		0.919	0.986	0.966	0.966					
		0.961	0.986	0.871	0.933					
		0.977	0.991	0.981	0.985					
ANN	0.959	0.958	0.981	0.958	0.959	34	n.c.	n.c.	n.c.	0.590
		0.949	0.993	0.982	0.972					
		0.906	0.989	0.906	0.955					
		0.988	0.990	0.977	0.987					

LDA—linear discriminant analysis, QDA—quadratic linear discriminant analysis, PLS-DA—partial least square discriminant analysis, SVM—support vector machine, RF—random forest, kNN—k-nearest neighbor, ANN—artificial neuronal network. Listed as barn, free-range, colony cage, and organic housing system; ^1^ AUC—area under the ROC (receiver operating characteristic) curve; ^2^ NMC—number of misclassifications; n.c.—not calculable.

**Table 3 foods-13-01098-t003:** Results of the purchased sample set of each computed model. More details can be found in Appendix A.

Samples	Number of Eggs	Housing System (Labeled)	Number of Eggs Classified as Labeled						
			LDA	QDA	PLS-DA	SVM	RF	kNN	ANN
V-001	12	organic	12	12	11	12	12	12	12
V-002	12	free-range	12	12	10	12	11	12	12
V-003	12	organic	11	12	9	12	2	5	10
V-004	10	free-range	10	10	10	10	4	8	10
V-005	10	organic	9	8	10	7	10	5	8
V-006	10	barn	0	0	8	0	0	6	2
V-007	10	organic	10	10	7	10	7	2	9
V-008	10	free-range	6	0	0	4	5	3	6
V-009	10	barn	1	0	3	0	3	0	0
V-010	10	barn	1	0	3	0	10	10	0
V-011	12	organic	12	12	0	12	10	4	2
V-012	12	organic	10	11	5	12	2	4	7
V-013	12	free-range	12	12	12	12	11	11	12
V-014	10	barn	0	0	2	0	0	0	0
V-015	9	free-range	9	5	7	8	0	1	8
V-016	10	barn	1	1	3	0	7	0	1
V-017	9	organic	9	9	9	9	4	6	9
V-018	9	organic	8	9	9	9	8	9	9
V-019	9	free-range	1	5	1	0	0	3	3
V-020	10	barn	7	0	3	0	3	2	4
V-021	10	barn	9	0	9	2	10	10	4
V-022	10	free-range	0	7	0	0	5	5	3
V-023	10	free-range	1	0	4	0	0	4	9
V-024	10	barn	9	0	8	2	4	1	7
V-025	10	free-range	1	8	0	0	0	1	4
V-026	12	organic	12	12	11	12	7	5	11
V-027	10	free-range	9	10	3	8	6	4	9
V-028	10	barn	0	0	3	0	8	0	0

## Data Availability

The data presented in this study are available on request from the corresponding author. The data are not publicly available due to privacy restrictions.

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
