# Peer review of "Authentication of Laying Hen Housing Systems Based on Egg Yolk Using 1H NMR Spectroscopy and Machine Learning"

_foods, 2024, doi:10.3390/foods13071098_

Round 1

Reviewer 1 Report

Comments and Suggestions for Authors

Dear Authors,

I have carefully reviewed your manuscript entitled "Authentication of laying hen housing systems based on egg yolk using 1H NMR spectroscopy and machine learning" and would like to offer the following comments for your consideration:

**Grammar and Language Issues:**

1. The phrase "is a target of food fraud due to the possibility of easy mislabeling of eggs" could be more concisely stated as "is susceptible to food fraud due to the potential for egg mislabeling."

2. The sentence "Yolks of 4,371 eggs from laying hens of four different breeds from colony cage, barn, free-range and organic housing systems were extracted and measured by 1H NMR spectroscopy" could be improved for clarity by rephrasing to "A total of 4,371 egg yolks, obtained from four different breeds of laying hens housed in colony cage, barn, free-range, and organic systems, were analyzed using 1H NMR spectroscopy."

3. In the sentence "The comparison of the seven computed models showed that the Support Vector Machine (SVM) model gives the best results with an accuracy of cross-validation of 98.5 %," the verb tense should be corrected to "gave" to maintain consistency.

4. The sentence "Eggs were collected throughout the entire laying period of the hens (20 to 94 weeks)" lacks clarity regarding the sampling strategy; it should specify if eggs were collected at regular intervals or at random.

**Experimental Design Concerns:**

1. In the "2.1 Raw Material" section, the description of the sample sources is not clearly articulated. It is suggested that a detailed table be provided to specify the origins of your samples, as well as the subsequent datasets used for training or prediction.

2. In the "Sample Preparation" section, since the authors have introduced several preprocessing steps, it is recommended to create a flowchart of the experimental procedure to illustrate the process of the experimental operations.

**Figure and Imagery Concerns:**

The manuscript refers to Figure 1 but does not provide a descriptive legend or explanation for the figure, which is essential for understanding the presented data.

The manuscript lacks visual aids such as figures or tables that could help elucidate the experimental design, data analysis, or results interpretation.

What were the results of the egg label prediction using supermarket eggs for grading by the authors? Were eggs from the same batch predicted to have more than one label? For the recognition of unknown samples, should single-class classifiers, such as SVDD, SIMCA, etc., be adopted?

I recommend that the authors address these points in a revised manuscript to enhance the clarity, rigor, and applicability of the research presented.

Best regards,

Reviewer 2 Report

Comments and Suggestions for Authors

The paper presents a novel approach that combines 1H NMR spectroscopy with machine learning to verify the housing systems of laying hens using egg yolk. This method tackles a substantial problem in food loss by offering a unique alternative to guarantee the genuineness of eggs. An impressive aspect of this study is the evaluation of seven machine learning models to identify the most efficient one for categorizing egg yolks based on their housing systems. Although authors have tried to explain their results yet, the manuscript has some issues, and it should be revised.

Improvements required: -

  1. (Lines 8 - 20): Abstract is very poor; it is highly unconventional and seems like point wise description of the manuscript architecture. Please edit abstract which look more academic.
  2. Graphical abstract: If possible, please add graphical abstract of the study, which represents the importance of the study.
  3. (Lines 65-74): In the section of “Raw material” please provide further details in the description of the sample collection and preparation for 1H NMR analysis would improve reproducibility. It would be beneficial to provide clear guidelines for selecting housing systems and managing samples, such as the conditions were similar or different for every collected eggs.
  4. (Lines 105-113): Please elaborate how ML will tackle the unbalanced data; include additional details about the validation procedure for the machine learning models, particularly regarding their handling of unbalanced data, a common occurrence in multi-class classification issues.
  5. (Lines 146-150): Please add few lines here so that the study would be enhanced by doing a more comprehensive comparative analysis with the existing methods and demonstrating the advantages of the suggested strategy above others would be beneficial.
  6. (Lines 248-256): The authors acknowledge the restriction pertaining to the representativeness of the sample size in relation to the overall population of laying hens in Germany. Please elaborate this constraint by adding few lines with respect to potential effects on the study's generalizability would enhance the manuscript.
  7. (Lines 260-266): While the influence of the breed on egg metabolome is mentioned by authors, however, the discussion on how this factor was controlled or its potential effects on the classification models is insufficient. Please add few lines explaining more in-depth analysis or discussion on this aspect which could eventually improve the manuscript.
  8. (Lines 105-113): To enhance the manuscript, it is advisable to include additional details about the validation procedure for the machine learning models, particularly regarding their handling of unbalanced data, a common occurrence in multi-class classification issues.

(Lines 276-287): Please add some lines in the discussion of SVM model; The scope of the discussion might be broadened to encompass further analysis of the practical ramifications of deploying the SVM model in real-life scenarios, such as its regulatory or industrial uses, as well as the potential obstacles that may arise. While future research directions are briefly mentioned, a more detailed discussion on specific next steps, potential improvements to the methodology, and other related applications of the research would be beneficial.

Comments on the Quality of English Language

Line 132-133: check for comma and syntax error, similarly, check line 10, 198, 207, 273

Check for grammatical errors
